# Machine Learning Model Based on Optimized Radiomics Feature from ^18^F-FDG-PET/CT and Clinical Characteristics Predicts Prognosis of Multiple Myeloma: A Preliminary Study

**DOI:** 10.3390/jcm12062280

**Published:** 2023-03-15

**Authors:** Beiwen Ni, Gan Huang, Honghui Huang, Ting Wang, Xiaofeng Han, Lijing Shen, Yumei Chen, Jian Hou

**Affiliations:** 1Department of Hematology, Renji Hospital, School of Medicine, Shanghai Jiao Tong University, No. 160 Pujian Road, Shanghai 200127, China; 2Department of Nuclear Medicine, Renji Hospital, School of Medicine, Shanghai Jiao Tong University, No. 160 Pujian Road, Shanghai 200127, China

**Keywords:** multiple myeloma, ^18^F-FDG-PET/CT, radiomics features, prognostic value

## Abstract

Objects: To evaluate the prognostic value of radiomics features extracted from ^18^F-FDG-PET/CT images and integrated with clinical characteristics and conventional PET/CT metrics in newly diagnosed multiple myeloma (NDMM) patients. Methods: We retrospectively reviewed baseline clinical information and ^18^F-FDG-PET/CT imaging data of MM patients with ^18^F-FDG-PET/CT. Multivariate Cox regression models involving different combinations were constructed, and stepwise regression was performed: (1) radiomics features of PET/CT alone (Rad Model); (2) Using clinical data (including clinical/laboratory parameters and conventional PET/CT metrics) only (Cli Model); (3) Combination radiomics features and clinical data (Cli-Rad Model). Model performance was evaluated by C-index and Net Reclassification Index (NRI). Results: Ninety-eight patients with NDMM who underwent ^18^F-FDG-PET/CT between 2014 and 2019 were included in this study. Combining radiomics features from PET/CT with clinical data showed higher prognostic performance than models with radiomics features or clinical data alone (C-index 0.790 vs. 0.675 vs. 0.736 in training cohort; 0.698 vs. 0.651 vs. 0.563 in validation cohort; AUC 0.761, sensitivity 56.7%, specificity 85.7%, *p* < 0.05 in training cohort and AUC 0.650, sensitivity 80.0%, specificity78.6%, *p* < 0.05 in validation cohort) When clinical data was combined with radiomics, an increase in the performance of the model was observed (NRI > 0). Conclusions: Radiomics features extracted from the PET and CT components of baseline ^18^F-FDG-PET/CT images may become an effective complement to provide prognostic information; therefore, radiomics features combined with clinical characteristic may provide clinical value for MM prognosis prediction.

## 1. Introduction

Multiple myeloma (MM), the second-most frequent hematologic tumor, is an incurable malignancy of the plasma cells. Over the past decade, the prognosis of MM has notably improved, due to the emergence of new therapeutic options. However, the improvement has not been uniform, and 15% to 20% of all patients have a predicted OS of less than 3 years [1]. Early identification of patients with high-risk features is needed to develop individualized and risk-adapted treatment strategies in newly diagnosed MM. Currently, several prognostic models have been used to stratify myeloma patients into subgroups with distinct risk profiles [2,3,4]. However, the performance of these models for identifying high-risk MM is not satisfactory.

^18^F-FDG PET/CT (18F-fluoro-deoxy-glucose positron emission tomography/computed tomography) is a useful diagnostic imaging procedure providing both tomographic and functional information in patients with MM. It may be regarded as a useful tool in the workup at diagnosis parameters and the follow-up of MM, especially for the detection of para-medullary and extramedullary disease or solid organ involvement. Various studies have demonstrated image-based standardized uptake value (SUV), extramedullary disease (EMD), and numbers of focal bone lesions (FLs) have been served as prognostic factors [5,6,7,8].

^18^F-FDG PET/CT was recommended by IMWG as the actual “gold standard” method for evaluating and monitoring response to anti-myeloma therapy [9]. As MM is highly heterogeneous, quantitative description of inter-tumoral and intra-tumoral heterogeneity might have significant potential for improved prognosis in MM. Consequently, it is necessary to develop more effective and feasible methods to assist in image analysis and mining more valuable prognostic information. Radiomics is an emerging area that shows promising prospects in the domain of radiological evaluation. Radiomics is a sophisticated image analysis technique that captures tissue and lesion high-throughput characteristics providing complementary information about tumor heterogeneity across the entire tumor volume to improve prognosis prediction and may therefore prove useful for patient stratification [10]. Increasing studies are published owning to encouraging results of radiomics-based machine-learning models. Most of these studies showed the value of radiomics extracted from PET was for solid tumor, such as lung cancer, head and neck cancer, and gastric cancer [11,12,13]. A recent study did clarify that the radiomics features model may predict high-risk cytogenetic status in multiple myeloma based on magnetic resonance imaging [14]. Some studies demonstrated that the radiomic analysis on standard CT or ^18^F-FDG-PET/CT images of patients with MM strongly improve accuracy in differentiating focal from diffuse patterns at diagnosis [15]. It also showed the value in disease follow-up, treatment options, and prognosis prediction. Bone marrow radiomics features extracted from ^18^F-FDG PET/CT may provide some information of MRD [16]. In a small sample size study, radiomics models based on MRI could also predict the response to bortezomib-based therapy in MM patients [17]. MRI-based textural features proved to correlate well with the clinical and hematological response (CR, VPGR, and PR) in MM patients undergoing systemic treatment [18]. In some sense, a radiomics approach may extract and mine more medical imaging features as reliable prognosis biomarkers of MM. We hypothesized that a model incorporating radiomic features extracted from baseline PET/CT would improve the prediction outcome of MM.

Although radiomics and machine learning have been widely used in disease diagnosis, the application of radiomics and multiple machine learning algorithms combined in predicting prognosis of MM has rarely been reported. The aim of this study was to evaluate the prognostic value of a machine learning model based on optimized radiomics features from ^18^F-FDG-PET/CT and clinical characteristics in NDMM patients.

## 2. Methods

### 2.1. Study Design and PATIENTS

We retrospectively reviewed medical records of 98 NDMM patients who underwent ^18^F-FDG-PET/CT between 2014 and 2019 in Renji hospital. Inclusion criteria included active MM, age of ≥18 years at the time of diagnosis, and availability of a pre-chemotherapy PET-CT scan and complete clinical data. Patients with a history of other tumors were excluded. This retrospective study was approved by institutional ethics committee in our hospital, and the informed consent requirement was waived. 

### 2.2. Data Collection

Baseline features of patients were used to characterize the disease at the beginning of the concerned period. We gathered initial results of PET/CT and the biomarkers performed before treatment in order to analyze the correlation between these characteristics and the prognosis of myeloma. The data set is divided into training set and test set by date at 70:30.

### 2.3. PET/CT Image Acquisition

According to the guidelines of the European Association of nuclear medicine (EANM), all patients underwent whole-body ^18^F-FDG positron emission tomography on Siemens Biograph-64 mCT scanner. All patients fasted for at least 6 h before acquisition, and the blood glucose levels were controlled below 150mg/dL. FDG-PET/CT was performed 60 min (60 ± 3 min) after injection of 3.7–5.55 MBq ^18^F-FDG per kg of body weight. PET image reconstruction with a 3-dimensional (3D) ordered-subset expectation maximization (OSEM) algorithm: 3 iterations, 24 subsets; 2.75 mm × 3.12 mm× 3.12 mm voxel size. The field of view (FOV) was 700 mm. Before PET scanning, CT was performed with attenuation correction methods to obtain image with matrix size of 512 × 512 (80 Ma, 120 kV). PET and CT results were reviewed on the workstation to display the fused image frame by frame. Then, the positron emission tomography image (voxel size 3.12 mm, slice thickness 2.75 mm) was interpolated to the same resolution as the computed tomography image (voxel size 0.98 mm, slice thickness 2 mm) (Appendix A Appendix A).

### 2.4. Image Preprocessing

^18^F-FDG-PET/CT images were read and interpreted by two independent board-certified nuclear medicine physicians with more than 10 years of experience. The osteolytic lesions are identified with a PET standard spatial resolution limit of about 5 mm. The maximum standardized uptake value (SUVmax) of the lesions obtained from the region of interest (ROI) is the standard semi-quantitative index that can be considered for image interpretation. If there is no focal FDG metabolism in visual analysis, the ROI with diameter of 10 mm is drawn at the first sacral vertebrae to obtain SUVmax. Focal lesions (FLs) at diagnosis were defined as focally increased FDG uptake greater than the physiologic bone marrow or liver uptake on at least two consecutive slices, with or without any underlying lytic lesion. The dichotomized number of FLs were with the threshold set at 3.

### 2.5. Radiomics Features Extraction and Selection

Skeleton volume of Interest (VOI) segmentation was mainly based on Slicer Radiomics (V2.10, https://github.com/Radiomics/SlicerRadiomics, accessed on 12 March 2022) as 3D Slicer extension which enables processing and extraction of radiomics features. To ensure the repeatability of PET/CT image features, we used the fixed bin width to acquire gray histogram and discrete image gray level. Finally, a total of 1702 image radiomics features were extracted from the original images of PET and CT by wavelet filter, including 18 first-order features, 13 shape features, 23 gray-level co-occurrence matrix features (GLCM), 16 gray-level scale matrix feature (GLSM), 16 gray-level size zone matrix (GLSZM), 5 neighborhood gray-tone difference matrix (NGTDM), and 14 gray-level dependence matrix (GLDM). The workflow was shown in Figure 1. All radiomics features were extracted from VOIs of PET and CT images. 

### 2.6. Predictive Model Establishment and Statistical Analysis

The data set is divided into training set and test set by date, and the proportion is 70:30, with the latest 30% used as the test-set. Optimal features are screened by univariate Cox regression together with least absolute shrinkage and selection operator (Lasso) algorithm and 10-fold cross-validation [19] (Figure 1 and Figure 2). Thus, different combinations were constructed, and stepwise regression was performed: (1) radiomics features of PET/CT alone (Rad-Mod); (2) using clinical data (including clinical/laboratory parameters and conventional PET/CT metrics) only (Cli-Mod); (3) combination radiomics features and clinical data (Cli-Rad-Mod). Receiver operating characteristic (ROC) curves were used to test the predictive performance of each model. The discriminative ability of each model was assessed by the concordance index (C-index). In order to evaluate the improvement in prediction performance gained by adding radiomics features to the baseline model, we calculated the net reclassification index (NRI) in the training cohort and validation cohort in the first and third year.

SPSS Statistics 26.0 (version 26.0; IBMC, Armonk, NY, USA) and R software packages (version 3.6.3, http://www.r-project.org, accessed on 10 February 2022) were used for statistical analysis and model construction. The Mann–Whitney U test and Chi-square test were used for comparisons between groups for continuous variables and categorical variables. Progression-free survival (PFS) was calculated from the beginning of treatment until disease progression or death from any cause. PFS were evaluated using Kaplan–Meier estimates. We used the Cox regression model to confirm the independent predictors of survival by univariate and multivariate analyses (see in Figure 2). The relative risk of an event and the 95% confidence interval (CI) were estimated using a Cox proportional hazard model. *p* < 0.05 indicates that the difference is statistically significant. The significant difference between two C-indices was tested using the Hmisc R package.

## 3. Result

### 3.1. Baseline Clinical Characteristics

The baseline clinical characteristics of the 98 patients are summarized in Table 1. The median age of all patients was 65 years (range, 41–86 years). The most prevalent type of MM patients was IgG type (49.0%), and the proportion of patients with light chain disease was 17.3%. The consensus of the International Myeloma Working Group defines high-risk multiple myeloma based on cytogenetics as having poor prognosis due to t (4; 14), del (17/17p), t (14; 16), t (14; 20), non-hyperdiploidy and gain (1q) [20]. In our study, cytogenetic abnormalities were investigated by FISH in 79 patients. Twenty-six (32.9%) MM patients were detected to have high-risk cytogenetic abnormalities. Overall, 62 patients (63.3%) received a treatment regimen containing a proteasome inhibitor as the first-line chemotherapy. Seventeen patients (17.3%) were treated with VRD regimen (bortezomib, lenalidomide, and dexamethasone). Fourteen patients (14.3%) were treated with daratumumab, melphalan, and dexamethasone. Twenty-two patients (22.4%) underwent consolidative autologous hematopoietic stem cell transplantation after induction chemotherapy. Among the 98 patients, 50 (51.0%) patients had more than three FLs. The median SUVmax in patients was 3.55 (range, 1.2–28.3), and the SUVmax in 41.8% patients was >4.2. Over the median follow-up of 27 (2.7–63) months, 52 patients (53.0%) had progressed and 22 (22.4%) had died. The median PFS was 28 months (95% CI, 20.5–35.5 months), and the median OS was 59 months (95% CI, 32.5–85.5 months). Patients with SUVmax greater than 4.2 were significantly associated with poorer survival than those who were with SUVmax lower than 4.2 (PFS, 14 vs. 40 months, *p* < 0.001). Patients who had more than three FLs were associated with significantly inferior PFS value compared with others (PFS, 20.0 vs. 38 months, *p* < 0.05) (Table 2).

### 3.2. Feature Selection and Model Performance

In this study, radiomics analysis showed a total of 1702 features were extracted, including morphological features, intensity features, texture features, and high-order features based on wavelet filters. Optimal radiomics features are screened by Lasso algorithm including LHL_Idmn_glcm, LHL_LDLGLE_gldm, LHL_LALGLE_glszm from (Table 3) retained as prognostic factors for models involving radiomics features. For models involving clinical parameters, elevated LDH (HR 1.004, 95% CI 1.000–1.008, *p* = 0.034) and SUVmax > 4.2(HR 1.114, 95%CI 1.043–1.189, *p* = 0.001) were consistently found to be significant predictors. After weighting the selected features according to the regression coefficient, the score of each patient were calculated, respectively. The highest Youden index was adapted from a time-dependent ROC curve used to determine the optimal cut-off value of each model. Patients were divided into high-risk group and low-risk group according to cut off value. The nomogram was constructed based on the above independent prognostic factors (Figure 3).

In this study, the model performance was evaluated by the concordance index (C-index). The value of C-index ranges from 0.5 to 1. The higher the c-index, the more accurate is the prediction. The C-index for each model is listed in Table 4. The C-index ranges of models with clinical data (including clinical/laboratory parameters and conventional PET/CT metrics) or radiomics features alone are 0.736 vs. 0.675 and 0.563 vs. 0.698 for the training and validation cohorts, respectively. Combination of clinical data and radiomics features showed higher C-index compared with models with clinical data (training cohort: C-index 0.790 [95% CI: 0.560–1.442] vs. 0.736 [95% CI: 0.401–1.600; validation cohort: 0.698 [95% CI: −0.346–1.048] vs. 0.563 [95% CI: −0.641–1.021]). Kaplan–Meier curves of PFS rates of each model in the subset are shown in Figure 4.

NRI is a measure for improvements in risk predictions. Table 5 summarizes the NRI in validation cohort for the first year and third year results for each combination. When clinical data were combined with radiomics, an increase in the performance of the model was observed. Adding radiomics features to the clinical model, the NRI was 0.482 (95% CI, −0.142 to 1.131) for the first year and 0.497 (95% CI, 0.142 to 1.131) for the third year. The NRI was 0.739 (95% CI, 0.350 to 1.380) for the first year and 0.632 (95% CI, 0.360 to 1.133) by adding clinical data to the radiomics model.

Table 6 and Figure 4 summarize the results for the AUC (area under ROC curve) of each combination. In comparison with the AUC for the clinical model, the significant improvement was seen with the combination of the clinical data and radiomics feature (*p* < 0.05). Cli-Rad model yielded the best performance (AUC 0.761, sensitivity 56.7%, specificity 85.7%, *p* < 0.05 in training cohort and AUC 0.650, sensitivity 80.0%, specificity 78.6%, *p* < 0.05 in validation cohort).

## 4. Discussion

MM is a condition that has a heterogeneous presentation and prognosis with survival rates ranging from months to decades. A prior identification of those with high-risk profiles is important for prognostication and personalized treatment strategies [21,22]. An increasing number of clinical prognostic markers for MM reflecting various aspects of the patients’ clinical status and disease behavior have been mentioned in the literature [23]; however, risk stratification is still a challenge because of spatial intra-tumoral heterogeneity. The imaging phenotype potentially containing extensive information of tumor characteristics and susceptibility to treatments can be partly acquired through medical image analysis, especially using PET-based images [24]. FDG-PET/CT enables detecting the presence of sites of metabolically active PCs and to assess changes in tumor cell metabolism after induction treatment. This study evaluated the potential prognostic performance of radiomics features extracted from FDG-PET/CT in MM integrated with clinical data. We have identified a model for predicting progression in newly diagnosed MM. Among the 13 clinical features initially considered in this study, LDH and SUVmax were selected in the final model. Patients with elevated LDH and SUVmax > 4.2 had significantly worse PFS. In this study, optimal radiomics features are screened including LHL_Idmn_glcm, LHL_LDLGLE_gldm, LHL_LALGLE_glszm from PET. Idmn is a measurement for local homogeneity of imagine. LALGLE reflects the proportion of a larger area with lower gray value in the image. LALGLE is a large area low gray level emphasis. Our model incorporated six of the most highly predictive PET/CT radiomics and clinical parameters. The model combining clinical data with radiomics features showed higher C-index than the models with clinical data alone (training cohort: C-index 0.790 [95% CI: 0.560–1.442] vs. 0.736 [95% CI: 0.401–1.600; validation cohort: 0.698 [95% CI: −0.346–1.048] vs. 0.563 [95% CI: −0.641–1.021]). In comparison with the AUC for the clinical model, the significant improvement was seen with combination of the clinical data and radiomics feature (*p* < 0.05). The Cli-Rad model yielded the best performance (AUC 0.761, sensitivity 56.7%, specificity 85.7%, *p* < 0.05 in training cohort and AUC 0.650, sensitivity 80.0%, specificity 78.6%, *p* < 0.05 in validation cohort).

Radiomics as a data-driven analysis of radiologic images might enable efficient mine image features providing valuable clinical information. Yet, few studies underly the interest of the value of radiomics features in MM. Radiomics features may quantify structural characteristics of bone marrow changes in MRI images and may be implemented as a complementary prognosis evaluation tool [25]. Some studies have shown MRI-based or PET/CT-based radiomics features may provide valuable information for image-based assessment of MRD and prediction of the therapy response [16,17,18,26]. Jamet B [27] tried to evaluate the potential prognostic value of textural features extracted from FDG-PET/CT in MM framework in addition to conventional PET-derived metabolic features and usual clinical/biological/genetic parameters. Though FDG-PET/CT has been considered a valuable tool in the work-up of patients with newly diagnosed MM, differentiation between focal and diffuse patterns on PET/CT is difficult. Therefore, some studies attempted to apply radiomic approaches to improve standard radiological evaluation with implications for prognosis. Tagliafico AS [28] found 15% of radiomics features (16/104) were different in diffuse and focal patterns. Mesguich C [21] found that a radiomic signature based on five different features extracted from PET and CT images was accurate for the diagnosis of diffuse disease in MM patients. In this study, we found radiomics features extracted from the baseline PET/CT combined with clinical parameters provided valuable information identifying the patients progressing early. Nevertheless, the limited literature could not give enough evidence of the value of radiomics features predicting outcomes in MM patients. A prior work on radiomics in myeloma has explicitly shown that the feature stability between different scanners is very limited in vivo, even after application of a simple image normalization. Radiomics features selected by a repeatability experiment only are not necessarily suited to build radiomics models for multicenter clinical application. Supposedly, one of the main reasons that hinder the translation to clinical application is the low external generalizability of radiomics models [29]. Accordingly, standardization of image acquisition or advanced calculative approaches for image normalization or RF compensation might help to improve external generalizability of radiomics prediction models. Further investigations that completely explore the potential prognostic value of PET/CT radiomics feature predicting the outcome of MM patients should be taken.

Our study has several limitations. First, our findings are based on a small size cohort from one institution with retrospective nature. A second follow-up duration may not be long enough; therefore, we establish a predictive model based on a single survival endpoint (PFS). Thus, a prospective multicenter study with a large cohort is necessary to confirm the results. Thirdly, the whole spine including the intervertebral disc was segmented in our study. The segmentation in a further study with using only the “clean” bone without discs might provide more valuable information on MM prognosis. Another research group combined the automatic BM segmentation with a subsequent radiomics analysis to automatically perform comprehensive, bone-by-bone phenotyping of the BM from wb-MR images which correctly exclude intervertebral discs [26]. This also brings inspiration for our future work.

## 5. Conclusions

Early identification of high-risk myeloma would help the development of precise treatment strategies. Radiomics features extracted from the baseline PET/CT quantitatively characterized intratumor heterogeneity and provided complementary information of prognosis for myeloma patients. In our study, the combination of radiomics features with clinical data showed improved performance relative to models with radiomics features or clinical parameters alone. Multivariate Cox model containing the radiomics information stratified patients into different risk groups for PFS, and thereby may mine more intratumor heterogeneous information and maybe further improve prognostic performance. Further studies with external test data will be needed to investigate the final, realistic performance of the model.

## Figures and Tables

**Figure 1 jcm-12-02280-f001:**
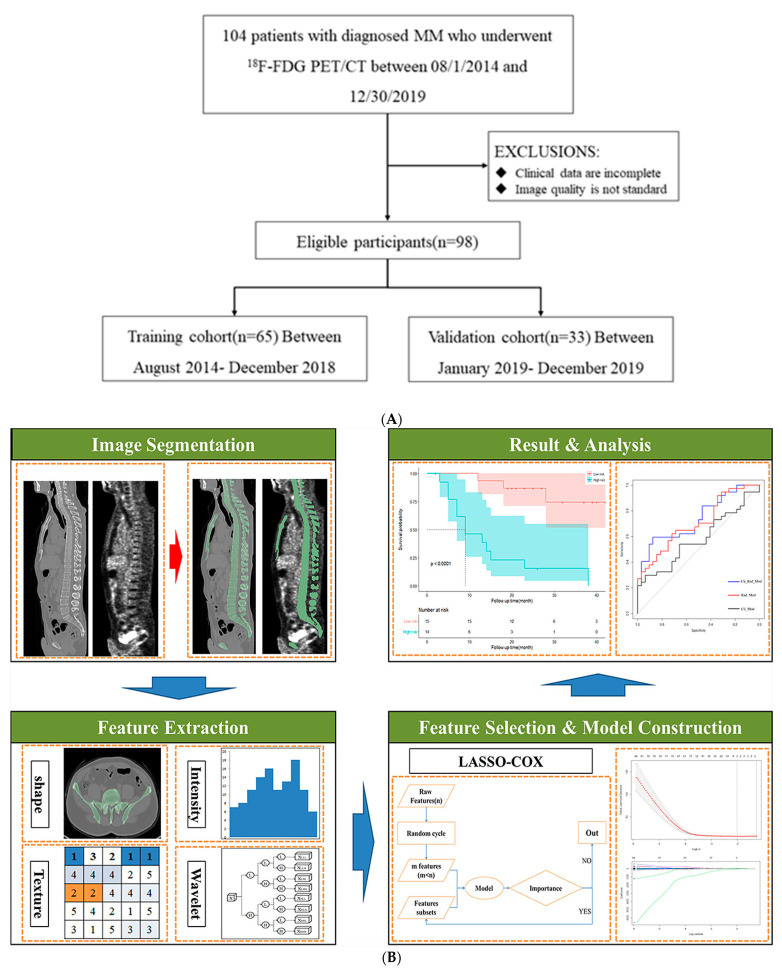
(**A**) Flowchart of the patient selection process. (**B**) Radiomics workflow. First, a region of interest is defined and/or lesions are segmented. A frequently large number of feature candidates are extracted. Optimal features are screened by Lasso algorithm and 10-fold cross-validation. Prognostic scores were generated for each multivariate Cox model by summing the product of each feature retained in the different models.

**Figure 2 jcm-12-02280-f002:**
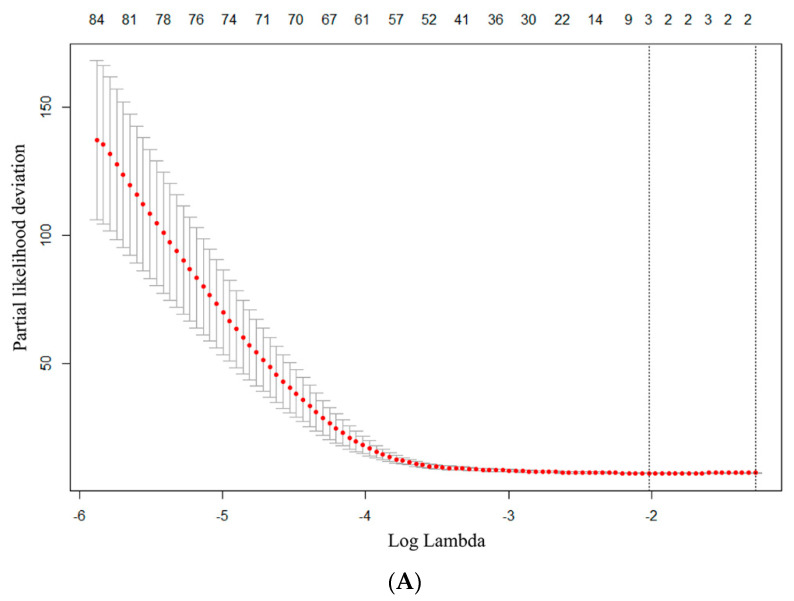
The radiomics features were screened based on lasso-Cox model. (**A**) The minimum criteria for 10-fold cross-validation was used to select optimal features. (**B**) Lasso coefficient analysis of 1702 radiomics features. Each curve in the graph represents the change track of each independent variable coefficient. With the increase of lambda, the coefficients of each feature are gradually compressed and tend to zero. The upper horizontal axis represents the characteristic number of radiomics, the lower horizontal axis represents the penalty coefficient log (lambda), the left vertical line represents the characteristic parameter values corresponding to the minimum partial likelihood deviation in the cross validation, and the right vertical line represents the corresponding parameter values within a standard error.

**Figure 3 jcm-12-02280-f003:**
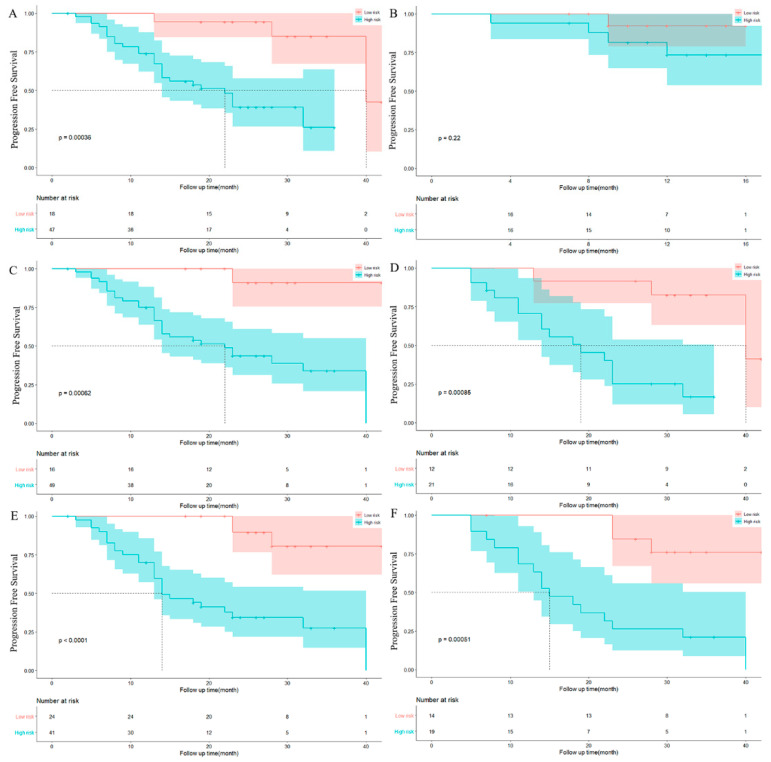
Kaplan–Meier Progression-free survival analysis of high and low risk group. (**A**,**B**) showed Kaplan–Meier PFS analysis of Cli-mod in the training and validation set; (**C**,**D**) showed Kaplan–Meier PFS analysis of Rad-mod in training and validation set; (**E**,**F**) showed Kaplan–Meier PFS analysis of Cli-Rad mod in training and validation set.

**Figure 4 jcm-12-02280-f004:**
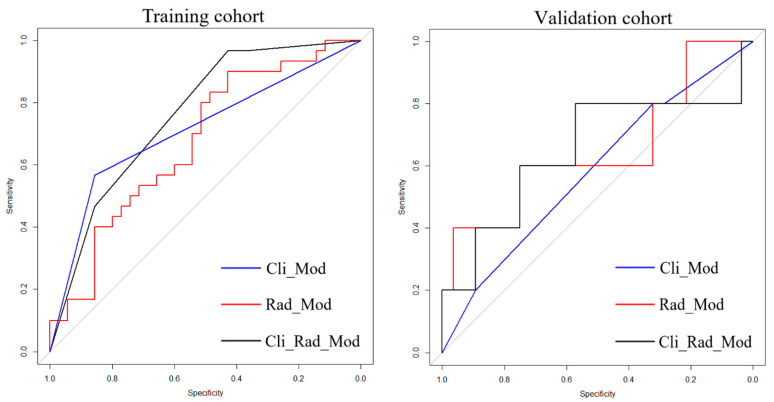
ROC (receiver operating characteristic curve) of different models in training and validation set.

**Table 1 jcm-12-02280-t001:** Baseline clinical characteristics of all patients (n = 98).

Variables	
Median age, years (range)	65.0 (41.0–86.0)
≥65 years, n (%)	52 (53.1%)
gender, n (%)	
Male	45 (45.9%)
Female	53 (54.1%)
Immunoglobulin (Ig) type, n (%)	
IgG	48 (49.0%)
IgA	25 (25.5%)
IgD	7 (7.1%)
Light chain only	18 (18.4%)
International Staging System (ISS), n (%)	
I	11 (11.2%)
II	49 (50.0%)
III	38 (38.8%)
ECOG PS ≥ 2, n (%)	22 (22.4%)
Bone marrow plasmacyte ratio (BMPC ratio) ≥ 60%	12 (12.3%)
Hemoglobin (g/L) < 100, n (%)	65 (66.3%)
LDH > (1 × ULN), n (%)	25 (25.5%)
β2MG (mg/L) ≥ 5.5	55 (56.1%)
Albumin (g/L) < 35, n (%)	58 (59.2%)
Calcium (mmol/L) > 2.65	12 (12.2%)
Creatinine (mg/dL) ≥ 2	32 (32.7%)
Cytogenetic abnormality (79/98)	
High risk	26 (32.9%)
Standard risk	53 (67.1%)
Frontline treatment, n (%)	
IMiD-based	5 (5.1%)
Proteasome inhibitor-based	62 (63.3%)
IMiD + proteasome inhibitor	17 (17.3%)
Daratumumab-based	14 (14.3%)
Performance of ASCT, n (%)	22 (22.4%)
Best response (92/98)	
CR (complete remission)	32 (34.8%)
Not reach CR	60 (65.2%)

**Table 2 jcm-12-02280-t002:** Univariate and multivariate Cox analysis for PFS in the training and validation cohorts for patients with MM.

Variable	Univariate Cox Regression		Multivariate Cox Regression	
	HR (95% CI)	*p*	HR (95% CI)	*p*
Gender	1.353 (0.483, 3.793)	0.565		
Age	1.025 (0.957, 1.097)	0.479		
ISS staging	7.169 (1.141, 45.025)	0.036	1.147 (0.665, 1.978)	0.621
BMPC ratio	1.000 (0.975, 1.025)	0.998		
β2MG (mg/L)	1.007 (0.998, 1.016)	0.113		
Albumin (g/L)	1.015 (0.966, 1.066)	0.560		
Calcium (mmol/L)	1.101 (0.380, 3.188)	0.860		
Creatinine (mg/dI)	0.984 (0.953, 1.015)	0.311		
High-risk cytogenetics	1.301 (0.257, 6.600)	0.750		
Ki-67	1.003 (0.978, 1.028)	0.844		
LDH	1.005 (1.001, 1.010)	0.028 *	1.004 (1.000, 1.008)	0.034 *
Bone destruction	0.188 (0.031, 1.162)	0.072		
Focal lesion	4.603 (1.140, 18.596)	0.032 *	1.976 (0.878, 4.447)	0.100
SUVmax	1.189 (1.077, 1.314)	0.001 *	1.114 (1.043, 1.189)	0.001 *

* *p* < 0.05. MM. Abbreviation: MM: multiple myeloma, PFS: Progression Free Survival, BMPC ratio: Bone marrow plasmacyte ratio, ISS: International Staging System, β2MG: β2 microglobulin, LDH: lactate dehydrogenase.

**Table 3 jcm-12-02280-t003:** Radiomics features from Lasso regression analysis.

Orde	Wavelet-Transformation	Imaging Parameter	Radiomics Feature	Feature Type
1	LHL	PET	Idmn	glcm
2	LHL	PET	LDLGLE	gldm
3	LHL	PET	LALGLE	glszm

**Table 4 jcm-12-02280-t004:** The C-index of each model in the training and validation cohorts.

Model	Training Cohort		Validation Cohort	
	C-index	(95% CI)	C-Index	(95% CI)
Cli_Mod	0.736	0.401, 1.600	0.563	−0.641, 1.021
Rad_Mod	0.675	0.376, 1.624	0.651	−0.597, 3.270
Cli_Rad_Mod	0.790	0.560, 1.442	0.698	−0.346, 1.048

**Table 5 jcm-12-02280-t005:** NRI in validation cohort for the first year and third year.

Model	Validation Cohort (1 Y) NRI (95% CI)	Validation Cohort (3 Y) NRI (95% CI)
Cli_Mod	Reference	Reference
Cli_Rad_Mod	0.482 (−0.149, 1.131)	0.497 (0.142, 1.131)
Rad_Mod	Reference	Reference
Cli_Rad_Mod	0.739 (0.350, 1.380)	0.623 (0.360, 1.133)

NRI > 0 indicates that the prediction ability of the new model is improved compared with the old mode (positive improvement); NRI < 0 indicates the prediction ability of the new model decreases (negative improvement); NRI = 0 is considered that the new model has not improved.

**Table 6 jcm-12-02280-t006:** Comparison of different models in training cohort and validation cohort.

	Training Cohort				Validation Cohort			
	AUC	SEN	SPE	*p* Value	AUC	SEN	SPE	*p* Value
Cli_Mod	0.692 (0.542, 0.804)	0.667	0.600	Reference	0.582 (0.305, 0.860)	0.600	0.821	Reference
Rad_Mod	0.673 (0.645, 0.877)	0.500	0.886	0.328	0.650 (0.350, 0.950)	0.600	0.786	0.058
Cli_Rad_Mod	0.761 (0.698, 0.906)	0.567	0.857	0.042 *	0.650 (0.339, 0.961)	0.800	0.786	0.036 *

* *p* < 0.05.

## Data Availability

The data presented in this study are available on request from the corresponding author. The data are not publicly available due to privacy or ethical restrictions.

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
