# Peer review of "Machine Learning Model Based on Optimized Radiomics Feature from 18F-FDG-PET/CT and Clinical Characteristics Predicts Prognosis of Multiple Myeloma: A Preliminary Study"

_jcm, 2023, doi:10.3390/jcm12062280_

Round 1

Reviewer 1 Report

In this work, Beiwen Ni and colleagues want to evaluate the prognostic role of radionics features derived from PET/CT images integrated with clinical characteristics and conventional PET/CT metrics in newly diagnosed MM patients.

I have, however, mixed feeling about the submitted paper for several reasons. Questions:

-       Why in 5 years (between 2014 and 2019) they reviewed medical records of only 98 patients? It is due to the inclusion criteria? Or image analysis was possible only in a low percentage of patients? In this regard, in what percentage of patients is the application of Slicer Radiomics possible?

-       Page 5, image 2 the left vertical line (partial likelihood deviation) is partially deleted;

-       Page 6, figure 3: It is not clear. I suppose the caption refers to the figure of page 7, it must be reformulated;

-       Page 7, caption of ROC curve (figure 5 at page 10?), it must be reformulated;

-       Page 10, table 6: what does the asterisk in the table refer to? must be made explicit in the table caption. moreover, at a clinical level, do you think the difference between Rad_Mod and Cli_Rad_Mod is significant?

I agree with the limitations of the study, first of all the small size cohort of a single center and the retrospective nature of this study, although at a clinical level I hope that the results of this study will be confirmed in prospective large cohort multicentric studies.

Author Response

Comment 1:Why in 5 years (between 2014 and 2019) they reviewed medical records of only 98 patients? It is due to the inclusion criteria? Or image analysis was possible only in a low percentage of patients? In this regard, in what percentage of patients is the application of Slicer Radiomics possible?

Response:Due to the high cost of PET/CT in 2014-2017, it was limited in clinical use. At present, PET/CT has been widely used in NDMM patients in our center due to its advantages in myeloma evaluation. Therefor we believe that radiomics approach may extract and mining more medical imaging features as reliable prognosis biomarkers of MM.

Comment 2: Page 5, image 2 the left vertical line (partial likelihood deviation) is partially deleted;

Response:Figure 2 showed Radiomics nomogram was used to predicte PFS in .1, 2 and 3 -year. SUVmax, LDH, Best response and rad-score are factors located on each axis. A straight line of each patient drawn up to the point axis of each factor. The points determined on the scale of each factor are summed to get a total score. In order to find the survival probability of patients in 1, 2 and 3 years, this line is drawn in the figure2.

Comment 3:   Page 6, figure 3: It is not clear. I suppose the caption refers to the figure of page 7, it must be reformulated;

-       Page 7, caption of ROC curve (figure 5 at page 10?), it must be reformulated;

Response: I think there were some mistakes in the number and caption of Figure 2, Figure 3 and Figure 4. It has been reformulated in the article. Figure 3 was Kaplan-Meier Progression-free survival analysis of high and low risk group. Figure 4 was ROC (receiver operating characteristic curve) of different models in training and validation set.

Comment 4:       Page 10, table 6: what does the asterisk in the table refer to? must be made explicit in the table caption. moreover, at a clinical level, do you think the difference between Rad_Mod and Cli_Rad_Mod is significant?

Response:

※ means that compared with Cli_ Mod, Cli_ Rad_ Mod and Cli_ Mod has significant difference. Rad_ Mod and Cli_ Rad_ Mod had no significant difference between the training group (p=0.203) and the validation group (p=0.160).

Reviewer 2 Report

The authors investigate the clinically very interesting question whether a radiomics approach based on PET-CT can improve risk stratification in patients with multiple myeloma. However, there are several aspects regarding the cohort, the methodological approach, study design, reporting, and the references, which need to be improved prior to publication:

Methods / Study Design:

L91: “Data set is randomly divided into training set and test set 91 at 7:3.”

The authors report a “random split”, however random splits are not encouraged. It is well known in the field of processing high-dimensional data and machine learning, that when splitting into training and testing data “randomly”, the investigators could easily test many different random splits until they find a split in which the model actually performs well, and this is prone to reporting highly overoptimistic results. When a random split is used, it can never be known whether the authors used such methods to produce overoptimistic results. I do not want to insinuate in any way that the authors might have done this, but this option should just be excluded by proper study design from the beginning. The split should be done by a clear criterion, for example by date (first 70% in training, remaining 30% in testing set); or better, developing models on data from one MRI scanner / one institution, and testing on a data from a second MRI scanner / second institution. If there were different scanners used in this study, the preference would be to split by scanners.

L119: “Volume of Interest (VOI) segmentation was mainly based on Slicer Radiomics (V2.10, https://github.com/Radiomics/SlicerRadiomics) as 3D Slicer extension which enables processing and extraction of radiomics features.” 

1.     It is not clear which structures were segmented: Was the whole skeleton segmented? Or were all focal lesions segmented? 

2.     Was all segmentation done manually? Manual segmentations, especially of whole skeletons or all focal lesions in myeloma patients, are extremely time consuming. This will hinder translation of such models into clinical practice, and this needs to be properly discussed. Recently, methods for automatic segmentation for patients with multiple myeloma have been presented, which now enable automatic segmentation and then automatic feature calculation for multiple myeloma patients (doi: 10.1097/RLI.0000000000000891 and 10.1097/RLI.0000000000000932). While I do not expect the authors to train such an automatic segmentation algorithm for the revision of this paper, these prior works need to be properly discussed.

L123: “1702 image radiomics features were extracted from the original images of PET and CT”. Does this mean 3404 features per patient in total?

These are very many features, especially when considering the very small, monocentric cohort with only 98 patients. Usually, the number of features should be approximately the same as the number of patients in the training data set; So I would recommend not to include any features from transformed images.

Model building:

-       Feature selection: was the feature selection done on all data or just on the training set?

-       It is unclear whether the authors trained several models on the training set and tested them all on the test-set, and then just reported the 3 best models (one for  radiomics features only and one for clinical features only and one for radiomics plus clinical features), or whether they performed all modelling only on the training data (by again splitting this data set in training and validation set), and kept the test set as an independent, “un-touched” hold-up test set.

-       L 210: “The highest Youden index was adapted from time-dependent ROC curve to determine the cut-off score of each patient.” Why is there an individual cutoff for each patient? There should be one cutoff for the model, and this should then be applied to stratify the test set.

-       L 262: Did the authors include best response category in the clinical model? I would have expected that only factors from baseline are included in the clinical model, to have a fair comparison between baseline PET-CT and other clinical baseline factors. The future response status is not known at baseline, so it does not make sense to build a model to predict survival at baseline, when the future response status can not yet be known at the time were the prognostic assessment is performed.

Results:

In Figure 1, the authors show that the whole spine including the intervertebral disc is segmented. Including the intervertebral discs might make the results worse, as it distorts the features which should only analyze the bone in myeloma, not the discs. It would be better to segment the spine without intervertebral disc as published before (doi:10.1097/RLI.0000000000000891). If this cannot be accomplished, it should be at least mentioned in the discussion that by improving the segmentation to using only the “clean” bone without discs might improve the results.

L217: “The C-index ranges of models with clinical data (including clinical/laboratory parameters and conventional PET/CT metrics) or radiomics features alone are 0.685 vs.0.721 and 0.728 vs.0.733 for training and validation cohorts, respectively.” 

It is extremely unlikely that the results are better on the validation set then in the training set on which the model was fitted on. How do the authors explain this?

Discussion:

The prior work by Jamet had a much larger data set and reported worse results in an independent test set. It is surprising that the current study showed better results, even though the data set was much smaller. This might be the results of overfitting and absence of in independent, external test set.

Missing external test set: Even after almost 1 decade of Radiomics, and despite the fact that every week approximately 50 new Radiomics papers appear on pubmed, Radiomics did not make it to large scale clinical practice yet. This gap between the success of radiomics in research and the missing success in clinical translation must be overcome, and addressing this must be an essential part of every new radiomics paper. The authors do address the fact that they did not  have an external test set to investigate the external generalizability in the limitations, but they further need to expand on this very important point. It had already been reported that bone (marrow) signal intensities can markedly deviate between different scanners (doi: 10.1097/RLI.0000000000000838), so the authors should consider using a normalization approach to ameliorate this problem at least in part, to pave the way for multicentric applicability of their algorithm.

More importantly than the signal intensity, it has recently been shown that in vivo, the reproducibility in radiomics features of the bone between different scanners is markedly below the repeatability of features ( 10.1097/RLI.0000000000000927), which finally provides hard evidence regarding the question why the performance of radiomics models markedly declines in external test sets. Therefore, it must be expected that the performance of the established models in a large-scale clinical application would be lower than the performance reported from the current study, and this must clearly be stated in the limitations. It should also be added to the conclusion that further studies with external test data will be needed to investigate the final, realistic performance of the model. Alternatively, an external test set could be added to investigate the performance in external institutions.

References:

L66/67: “Patients with diffuse disease belong to a high-risk category, with 66 frequent association with high-risk cytogenetics features [12]”: 

This citation is wrong. In this work, the authors claimed that they can predict cytogenetic high-risk status from MRI, based on a very small dataset without external, independent test set. There was no evidence in this study that especially diffuse disease is connected to a certain genetic risk group. 

The authors cite some more general radiomics studies, but as the current study uses Radiomics in MM, there are several missing citations on earlier works using radiomics in MM:

doi: 10.1016/j.cmpb.2022.107083

doi: 10.1097/RLI.0000000000000891

doi: 10.3389/fonc.2021.709813

doi: 10.1097/RCT.0000000000001298

doi: 10.1097/RLI.0000000000000927

doi: 10.1155/2022/6911246

doi: 10.3390/cancers12030761

L74: “Although radiomics and machine learning have been widely used in disease diagnosis, the application of radiomics and multiple machine learning algorithms combined in predicting prognosis of MM has rarely been reported.” References are missing.

The references must be improved, especially regarding the two central challenges for bringing radiomics from research to clinical practice, which are generalizability and workflow automation with automatic segmentation, as laid out above.

Reporting:

Inclusion criteria are not well defined: L84: “other biomarkers.”: which exact other markers needed to be present for inclusion?

Figure 4: It is not shown which graph shows which model, and this information is also not given in the legend.

Table 1: There are many different classifications of cytogenetic risk in Myeloma: To which does “cytogenetic abnormality” refer here? A reference needs to be provided.

There are many language problems throughout the manuscript which make it hard to understand what the authors did and how they interpret their findings, starting in the first line of the abstract. For example:

L 12: derided instead of derived

L 22: if the study was retrospective, patients were not recruited, but rather included.

L34: Hematooncologic entities as Myeloma are not tumors, they are malignancies.

L 62: radiomics-based patients: What does did mean?

Author Response

The authors investigate the clinically very interesting question whether a radiomics approach based on PET-CT can improve risk stratification in patients with multiple myeloma. However, there are several aspects regarding the cohort, the methodological approach, study design, reporting, and the references, which need to be improved prior to publication:

Methods / Study Design:

Comment 1: L91: “Data set is randomly divided into training set and test set 91 at 7:3.”

The authors report a “random split”, however random splits are not encouraged. It is well known in the field of processing high-dimensional data and machine learning, that when splitting into training and testing data “randomly”, the investigators could easily test many different random splits until they find a split in which the model actually performs well, and this is prone to reporting highly overoptimistic results. When a random split is used, it can never be known whether the authors used such methods to produce overoptimistic results. I do not want to insinuate in any way that the authors might have done this, but this option should just be excluded by proper study design from the beginning. The split should be done by a clear criterion, for example by date (first 70% in training, remaining 30% in testing set); or better, developing models on data from one MRI scanner / one institution, and testing on a data from a second MRI scanner / second institution. If there were different scanners used in this study, the preference would be to split by scanners.

Response: This study is based on machine learning model with small sample size. All images are collected from the same scanner. In the research design, grouping and training is mainly according to the classical machine learning method. The sample is randomly divided into training set and validation set at 7:3. In the subsequent model training, tenfold cross-validation is used to avoid subjective bias.

Comment 2: L119: “Volume of Interest (VOI) segmentation was mainly based on Slicer Radiomics (V2.10, https://github.com/Radiomics/SlicerRadiomics) as 3D Slicer extension which enables processing and extraction of radiomics features.”

  1. It is not clear which structures were segmented: Was the whole skeleton segmented? Or were all focal lesions segmented?
  2. Was all segmentation done manually? Manual segmentations, especially of whole skeletons or all focal lesions in myeloma patients, are extremely time consuming. This will hinder translation of such models into clinical practice, and this needs to be properly discussed. Recently, methods for automatic segmentation for patients with multiple myeloma have been presented, which now enable automatic segmentation and then automatic feature calculation for multiple myeloma patients (doi: 10.1097/RLI.0000000000000891 and 10.1097/RLI.0000000000000932). While I do not expect the authors to train such an automatic segmentation algorithm for the revision of this paper, these prior works need to be properly discussed.

Response: The extraction of the wholebody skeleton is based on the semi-automatic sketch of 3D slicer software. First, the skeleton is automatically recognized by setting an adaptive threshold, and then it is manually corrected. In particular, the focal lesions are finally determined by two independent board-certified nuclear medicine physicians with more than 10 years of experience. For the automatic segmentation of bone in patients with MM, we consider to improve training performance of the algorithm based on the increase of sample size.

Comment 3: L123: “1702 image radiomics features were extracted from the original images of PET and CT”. Does this mean 3404 features per patient in total?

Response:851 radiomics features were extracted from CT images and 851 features were extracted from PET images, with a total of 1702 features for each patient.

Comment 4: These are very many features, especially when considering the very small, monocentric cohort with only 98 patients. Usually, the number of features should be approximately the same as the number of patients in the training data set; So I would recommend not to include any features from transformed images.

Response:The radiomics features extracted in this study mainly include first-order features、shape features、texture features、high-older features, which have been widely used in study of radiomics. We used LASSO regression to eliminate redundant features effectively.

Model building:

Comment 5: Feature selection: was the feature selection done on all data or just on the training set?

Response: Feature selection was only performed on the training set.

Comment 6: It is unclear whether the authors trained several models on the training set and tested them all on the test-set, and then just reported the 3 best models (one for  radiomics features only and one for clinical features only and one for radiomics plus clinical features), or whether they performed all modelling only on the training data (by again splitting this data set in training and validation set), and kept the test set as an independent, “un-touched” hold-up test set.

Response: We trained three models in the training set. The first model used only radiomics features, the second model used only clinical features, and the third model used combination of clinical features and imaging features. These three models are verified on independent verification sets.

Comment 7: L 210: “The highest Youden index was adapted from time-dependent ROC curve to determine the cut-off score of each patient.” Why is there an individual cutoff for each patient? There should be one cutoff for the model, and this should then be applied to stratify the test set.

Response: This part has been modified and highlight in yellow. It should be based on the cut-off value of the ROC curve of each model as the cut-off point, and the cut-off point should be applied to stratify on the test set.

 Comment 8: L 262: Did the authors include best response category in the clinical model? I would have expected that only factors from baseline are included in the clinical model, to have a fair comparison between baseline PET-CT and other clinical baseline factors. The future response status is not known at baseline, so it does not make sense to build a model to predict survival at baseline, when the future response status can not yet be known at the time were the prognostic assessment is performed.

Response: The end point of this study was progression-free survival (PFS), so the best response to induction therapy was included in the clinical prognosis model.

Results:

Comment 9: In Figure 1, the authors show that the whole spine including the intervertebral disc is segmented. Including the intervertebral discs might make the results worse, as it distorts the features which should only analyze the bone in myeloma, not the discs. It would be better to segment the spine without intervertebral disc as published before (doi:10.1097/RLI.0000000000000891). If this cannot be accomplished, it should be at least mentioned in the discussion that by improving the segmentation to using only the “clean” bone without discs might improve the results.

Response:We have discussed the limitations of the segmentation method in our study.

Comment 10: L217: “The C-index ranges of models with clinical data (including clinical/laboratory parameters and conventional PET/CT metrics) or radiomics features alone are 0.685 vs.0.721 and 0.728 vs.0.733 for training and validation cohorts, respectively.”

It is extremely unlikely that the results are better on the validation set then in the training set on which the model was fitted on. How do the authors explain this?

Response: There were some errors in the data of the C-index that have been modified and high-light in yellow.

Discussion:

Comment 11: The prior work by Jamet had a much larger data set and reported worse results in an independent test set. It is surprising that the current study showed better results, even though the data set was much smaller. This might be the results of overfitting and absence of in independent, external test set.

Missing external test set: Even after almost 1 decade of Radiomics, and despite the fact that every week approximately 50 new Radiomics papers appear on pubmed, Radiomics did not make it to large scale clinical practice yet. This gap between the success of radiomics in research and the missing success in clinical translation must be overcome, and addressing this must be an essential part of every new radiomics paper. The authors do address the fact that they did not  have an external test set to investigate the external generalizability in the limitations, but they further need to expand on this very important point. It had already been reported that bone (marrow) signal intensities can markedly deviate between different scanners (doi: 10.1097/RLI.0000000000000838), so the authors should consider using a normalization approach to ameliorate this problem at least in part, to pave the way for multicentric applicability of their algorithm.

More importantly than the signal intensity, it has recently been shown that in vivo, the reproducibility in radiomics features of the bone between different scanners is markedly below the repeatability of features (10.1097/RLI.0000000000000927), which finally provides hard evidence regarding the question why the performance of radiomics models markedly declines in external test sets. Therefore, it must be expected that the performance of the established models in a large-scale clinical application would be lower than the performance reported from the current study, and this must clearly be stated in the limitations. It should also be added to the conclusion that further studies with external test data will be needed to investigate the final, realistic performance of the model. Alternatively, an external test set could be added to investigate the performance in external institutions.

Response: Thank you for your suggestion. As you suggested, further Study needs external verification to investigate the realistic performance of the model.

References:

Comment 12: L66/67: “Patients with diffuse disease belong to a high-risk category, with 66 frequent association with high-risk cytogenetics features [12]”:

This citation is wrong. In this work, the authors claimed that they can predict cytogenetic high-risk status from MRI, based on a very small dataset without external, independent test set. There was no evidence in this study that especially diffuse disease is connected to a certain genetic risk group.

Response: The above contents have been modified.

The authors cite some more general radiomics studies, but as the current study uses Radiomics in MM, there are several missing citations on earlier works using radiomics in MM:

doi: 10.1016/j.cmpb.2022.107083

doi: 10.1097/RLI.0000000000000891

doi: 10.3389/fonc.2021.709813

doi: 10.1097/RCT.0000000000001298

doi: 10.1097/RLI.0000000000000927

doi: 10.1155/2022/6911246

doi: 10.3390/cancers12030761

L74: “Although radiomics and machine learning have been widely used in disease diagnosis, the application of radiomics and multiple machine learning algorithms combined in predicting prognosis of MM has rarely been reported.” References are missing.

The references must be improved, especially regarding the two central challenges for bringing radiomics from research to clinical practice, which are generalizability and workflow automation with automatic segmentation, as laid out above.

Response:We sincerely appreciate the valuable comments. We have checked the literature carefully and added more references into the INTRODUCTION part in the revised manuscript.

Reporting:

Inclusion criteria are not well defined: L84: “other biomarkers.”: which exact other markers needed to be present for inclusion?

Response: Inclusion criteria has been modified.

Figure 4: It is not shown which graph shows which model, and this information is also not given in the legend.

Response: I think there were some mistakes in the number and caption of Figure 2, Figure 3 and Figure 4. It has been reformulated in the article. Figure 3 was Kaplan-Meier Progression-free survival analysis of high and low risk group. Figure 4 was ROC (receiver operating characteristic curve) of different models in training and validation set.

Table 1: There are many different classifications of cytogenetic risk in Myeloma: To which does “cytogenetic abnormality” refer here? A reference needs to be provided.

Response: According to the International Myeloma Working Group (IMWG) criteria, patients with del17p, t (4; 14), t (14;20) or t (14; 16) was defined as "high-risk MM".

There are many language problems throughout the manuscript which make it hard to understand what the authors did and how they interpret their findings, starting in the first line of the abstract. For example:

L 12: derided instead of derived

L 22: if the study was retrospective, patients were not recruited, but rather included.

L34: Hematooncologic entities as Myeloma are not tumors, they are malignancies.

L 62: radiomics-based patients: What does did mean?

Response: We tried our best to improve the manuscript and made some changes to the manuscript. These changes will not influence the content and framework of the paper. And here we did not list the changes but marked in red in the revised paper. We appreciate for Editors/Reviewers’ warm work earnestly and hope that the correction will meet with approval.

Reviewer 3 Report

Overall this is a fairly well written paper.

Please provide more data about cytogenetics. If possible, data about gain 1q should be included. 

Sample size is small. While this data is interesting, a larger sample size would be preferable for the cohorts.

Please consider rephrasing the conclusion. The data is not strong enough to support such the conclusion.

Author Response

Comment:Please provide more data about cytogenetics. If possible, data about gain 1q should be included. 

Response: In our study, patients with del17p, t (4; 14), t (14;20) or t (14; 16) was defined as "high-risk MM".

Comment: Sample size is small. While this data is interesting, a larger sample size would be preferable for the cohorts.

Response: Our findings are based on small size cohort from one institution with retrospective nature. Thus, prospective multicenter study with a large cohort is necessary to confirm the results. Further studies with external test data will be needed to investigate the final, realistic performance of the model.

Comment:Please consider rephrasing the conclusion. The data is not strong enough to support such the conclusion.

Response:The conclusion has been rephrased.

Round 2

Reviewer 2 Report

·      Former Comment 1: I have laid out why a random split is not acceptable, however the authors have not changed the split to a clear criterion, as 70:30 by date, with latest 30% used as test-set. Changing the data split is a requirement for consideration for acceptance of the article.

·      Former Comment 2: Segmentation: What do the authors mean by the skeleton is segmented automatically, and then the focal lesions are determined? It is still not clear whether the skeleton, or only the focal lesions, were segmented and used for the radiomics analysis. This is one of the most important informations for a radiomics study and needs to be clarified in the manuscript text.

·      Former Comment 8: It is not correct to include best induction therapy response as a clinical factor in the prognosis model, because this information is not present at baseline, when the radiomics analysis is performed. This is a major methodological error which needs to be corrected.

·      Former Comment 10: The authors have acknowledged that their former results were false, however they present new data in which again the performance on the validation set is better than in the training set. This is almost impossible, as the model in almost any case performs better on the data on which it was trained, than on new, independent test data. How do the authors explain this phenomenon?

·      Former Comment 11: the authors have not discussed any of the necessary points:

o   How do the authors explain that their results seem to be markedly better than the results from a much larger, former study (Jamet et al. Random survival forest to predict transplant-eligible newly diagnosed multiple myeloma 377 outcome including FDG-PET radiomics: a combined analysis of two independent prospective European trials. Eur J Nucl Med 378 Mol Imaging. 2021. 48(4): 1005-1015.) The most likely explanation is that the authors have used a very small data set, way too many radiomics features, and this has resulted in an overfitted model, with a seemly very good performance in this internal (random) test set, but which won´t generalize to external data sets. This needs to be clearly discussed, or an external, independent test-set needs to be added.

o   It has not been discussed that a prior work on radiomics in myeloma has explicitly shown that the feature-stability between different Scanners is very limited in vivo (10.1097/RLI.0000000000000927). This is one of the main reasons why this model is not expected to generalize well, and this fact also has to be properly discussed and cited.

·      Former Comment 9: It needs to be properly mentioned in the article that automatic segmentation algorithms for bone marrow of myeloma patients, which correctly exclude intervertebral discs, have already been developed by another group (doi:10.1097/RLI.0000000000000891).

·      Former Comment 12: The authors need to discuss the prior radiomics-studies performed in MM:

doi: 10.1016/j.cmpb.2022.107083

doi: 10.1097/RLI.0000000000000891 

doi: 10.3389/fonc.2021.709813

doi: 10.1097/RCT.0000000000001298 

doi: 10.1097/RLI.0000000000000927 

doi: 10.1155/2022/6911246

doi: 10.3390/cancers12030761

·      Definition of high-risk cytogenetics: The authors claim that patients with del17p, t (4; 14), t (14;20) or t (14; 16). This definition is wrong, t(14;20) is not included in the current high-risk definition. The authors need to correct this, and need to add a citation to which cytogenetic definition of the IMWG they are referring. 

Author Response

Dear reviewer:

Thank you for your decision and constructive comments on my manuscript. We have carefully considered the suggestion of Reviewer and have tried our best to improve and made some changes in the manuscript. As you commented, it is inappropriate to include ‘best induction therapy response’ as a clinical factor in the prognosis model, so we removed this factor from the prognosis model and data has been re-analyzed. Tables and figures have been modified.. The yellow part that has been revised according to your comments. Revision notes, point-to-point, are given as follows:

  Former Comment 1: I have laid out why a random split is not acceptable, however the authors have not changed the split to a clear criterion, as 70:30 by date, with latest 30% used as test-set. Changing the data split is a requirement for consideration for acceptance of the article.

Response: Considering the Reviewer’s suggestion, we have changed the random split to a clear criterion. The split has been done by date (first 70% in training, remaining 30% in testing set).

  • Former Comment 2: Segmentation: What do the authors mean by the skeleton is segmented automatically, and then the focal lesions are determined? It is still not clear whether the skeleton, or only the focal lesions, were segmented and used for the radiomics analysis. This is one of the most important informations for a radiomics study and needs to be clarified in the manuscript text.

Response: Focal lesions (FLs) at diagnosis were defined as focally increased FDG uptake greater than the physiologic bone marrow or liver uptake on at least two consecutive slices, with or without any underlying lytic lesion. It is a conventional PET/CT feature determined in visual analysis. The skeleton was segmented and used for the radiomics analysis.

  • Former Comment 8: It is not correct to include best induction therapy response as a clinical factor in the prognosis model, because this information is not present at baseline, when the radiomics analysis is performed. This is a major methodological error which needs to be corrected.

Response: As you commented, it is inappropriate to include best induction therapy response as a clinical factor in the prognosis model, so we have corrected this error. This factor is not included in the final prognosis model.

  • Former Comment 10: The authors have acknowledged that their former results were false, however they present new data in which again the performance on the validation set is better than in the training set. This is almost impossible, as the model in almost any case performs better on the data on which it was trained, than on new, independent test data. How do the authors explain this phenomenon?

Response:We removed the factor of ‘best response’ from the prognosis model and data has been re-analyzed. Tables and figures have been modified.

  • Former Comment 11: the authors have not discussed any of the necessary points:

How do the authors explain that their results seem to be markedly better than the results from a much larger, former study (Jamet et al. Random survival forest to predict transplant-eligible newly diagnosed multiple myeloma 377 outcome including FDG-PET radiomics: a combined analysis of two independent prospective European trials. Eur J Nucl Med 378 Mol Imaging. 2021. 48(4): 1005-1015.) The most likely explanation is that the authors have used a very small data set, way too many radiomics features, and this has resulted in an overfitted model, with a seemly very good performance in this internal (random) test set, but which won´t generalize to external data sets. This needs to be clearly discussed, or an external, independent test-set needs to be added.

Response:1)In the former study (2021. 48(4): 1005-1015), data was collected from two different trials(IMAJEM and EMN02/HO95). In the former study, given the numerous institutions involved and the small number of patients per institution, a harmonization process was used at the “country” level (France and Italy). The M-ComBat approach was implemented with the aim to transform values to a common reference chosen as the French one instead of a mean reference, which may lead to aberrant values. 2)Though TF were not retained in the final model predicting PFS, among the 7 other most predictive variables selected by the VIMP (RSF first approach), the parameter called ZLNU ranked second while other TF parameters like LGRE using the absolute quantization are also selected for some folds when using the nested cross-validation. These results can be partly explained by the huge and intelligible prognostic impact of the treatment arm in the cohort which could underestimate the prognostic value of other tested variables, especially TF. 3) Our study showed radiomics feature-stability between different Scanners is very limited in vivo. The images in our study obtained from the same scanner which ensured radiomics feature-stability. 4) We use Lasso regression, which is in common multiple linear regression, adding penalty function and constantly compressing coefficient to avoid collinearity and overfitting. Then, according to the 10-fold cross validation, we selected the optimal model. An external, independent test-set will be done in the further study.

  It has not been discussed that a prior work on radiomics in myeloma has explicitly shown that the feature-stability between different Scanners is very limited in vivo (10.1097/RLI.0000000000000927). This is one of the main reasons why this model is not expected to generalize well, and this fact also has to be properly discussed and cited.

Response: Thank you for your valuable comment, it is definitely a critical issue that the radiomics feature-stability between different Scanners is very limited in vivo. We have discussed this issue in the article and cited “standardization of image acquisition, or advanced calculative approaches for image normalization or RF compensation might help to improve external generalizability of radiomics prediction models.”

 Former Comment 9: It needs to be properly mentioned in the article that automatic segmentation algorithms for bone marrow of myeloma patients, which correctly exclude intervertebral discs, have already been developed by another group (doi:10.1097/RLI.0000000000000891).

Response: Thanks for Reviewer’s suggestion, we have mentioned the automatic segmentation algorithms for bone marrow of myeloma patients, which correctly exclude intervertebral discs in the discussion of our article.

  Former Comment 12: The authors need to discuss the prior radiomics-studies performed in MM:

doi: 10.1016/j.cmpb.2022.107083

doi: 10.1097/RLI.0000000000000891 

doi: 10.3389/fonc.2021.709813

doi: 10.1097/RCT.0000000000001298 

doi: 10.1097/RLI.0000000000000927 

doi: 10.1155/2022/6911246

doi: 10.3390/cancers12030761

Response: As Reviewer suggested that it is indeed better to give some reference for the prior radiomics-studies performed in MM. Seven reference were added to provide the findings of the studies of MRI-based radiomics in MM.

Comment: Definition of high-risk cytogenetics: The authors claim that patients with del17p, t (4; 14), t (14;20) or t (14; 16). This definition is wrong, t(14;20) is not included in the current high-risk definition. The authors need to correct this, and need to add a citation to which cytogenetic definition of the IMWG they are referring. 

Response: According the International Myeloma Working Group consensus in 2016, the definition for high-risk multiple myeloma based on cytogenetics is that s’everal cytogenetic abnormalities such as t (4;14), del(17/17p), t (14;16), t (14;20), non-hyperdiploidy and gain(1q) were identified that confer poor prognosis.

Reviewer 3 Report

Overall a decent article. Authors have made changes as recommended. There is more data regarding radiology evaluation in diagnosis and response assessment in multiple myeloma. 

Author Response

Thank you for your decision and constructive comments on my manuscript.